# Stimulus representation in human frontal cortex supports flexible control in working memory

Zhujun Shao[1,2†], Mengya Zhang[1†], Qing Yu[1*]

[1]Institute of Neuroscience, State Key Laboratory of Brain Cognition and Brain-inspired Intelligence Technology, Center for Excellence in Brain Science and Intelligence Technology, Chinese Academy of Sciences, Shanghai, China; [2]University of Chinese Academy of Sciences, Beijing, China

## eLife Assessment

This work presents **important** findings that the human frontal cortex is involved in a flexible, dual role in both maintaining information in short-term memory, and controlling this memory content to guide adaptive behavior and decisions. The evidence supporting the conclusions is **compelling**, with a well-designed task, best-practice decoding methods, and careful control analyses. The work will be of broad interest to cognitive neuroscience researchers working on working memory and cognitive control.

**\*For correspondence:**
qingyu@ion.ac.cn

[†]These authors contributed equally to this work

**Abstract** When holding visual information temporarily in working memory (WM), the neural representation of the memorandum is distributed across various cortical regions, including visual and frontal cortices. However, the role of stimulus representation in visual and frontal cortices during WM has been controversial. Here, we tested the hypothesis that stimulus representation persists in the frontal cortex to facilitate flexible control demands in WM. During functional MRI, participants flexibly switched between simple WM maintenance of visual stimulus or more complex rule-based categorization of maintained stimulus on a trial-by-trial basis. Our results demonstrated enhanced stimulus representation in the frontal cortex that tracked demands for active WM control and enhanced stimulus representation in the visual cortex that tracked demands for precise WM maintenance. This differential frontal stimulus representation traded off with the newly-generated category representation with varying control demands. Simulation using multi-module recurrent neural networks replicated human neural patterns when stimulus information was preserved for network readout. Altogether, these findings help reconcile the long-standing debate in WM research, and provide empirical and computational evidence that flexible stimulus representation in the frontal cortex during WM serves as a potential neural coding scheme to accommodate the ever-changing environment.

## Introduction

Real-world flexible behavior relies largely on WM, which allows the maintenance and manipulation of information in the brain in order to serve diverse behavioral goals (*Baddeley, 2003*). One central problem in the field of WM is to understand how stimulus information is represented and maintained in WM. Over the past decade, mounting evidence has demonstrated stimulus-specific representation during WM maintenance in a distributed cortical network, including sensory, parietal, and frontal cortices (*Christophel et al., 2012*; *Ester et al., 2015*; *Gosseries et al., 2018*; *Harrison and Tong,*

2009; *Riggall and Postle, 2012*; *Serences et al., 2009*; *Sprague and Serences, 2013*; *Yu and Shim, 2017*; *Yu and Shim, 2019*). However, the exact nature and functions of stimulus representation in different cortical regions remain controversial. Specifically, while neurophysiological studies in non-human primates have mostly emphasized stimulus representation in the frontal cortex (*Funahashi et al., 1989*; *Fuster and Alexander, 1971*; *Leavitt et al., 2017*), neuroimaging work in humans has reported disparate findings. During the maintenance of simple visual features, stimulus representation is robustly encoded in the early visual cortex (EVC), which has been taken as the evidence in support of the sensorimotor recruitment hypothesis of WM (*Harrison and Tong, 2009*; *Riggall and Postle, 2012*; *Serences et al., 2009*). Meanwhile, stimulus representation in the higher-order frontoparietal cortex is typically weaker and less stable (*Emrich et al., 2013*; *Gosseries et al., 2018*; *Riggall and Postle, 2012*; *Yu and Shim, 2019*). However, in dynamic environments such as those involving distraction, stimulus representation in EVC could be greatly interrupted or biased (*Bettencourt and Xu, 2016*; *Hallenbeck et al., 2021*; *Lorenc et al., 2018*). In contrast, stimulus representation in the frontal cortex could be robust under certain circumstances including attentional prioritization (*Christophel et al., 2018*), categorization (*Lee et al., 2013*), and after extensive training (*Miller et al., 2022*). To summarize, stimulus representation could vary markedly depending on specific brain regions and memory tasks, complicating the interpretation of potential functions of stimulus representation in WM.

In this study, we consider these apparent discrepancies from the perspective of cognitive flexibility (*Badre et al., 2021*; *Fusi et al., 2016*; *Musslick and Cohen, 2021*). We propose that changes in stimulus representation in different cortical regions might reflect a global reconfiguration in coding strategy and resource allocation in response to varied WM functions (*Henderson et al., 2022*; *Lee et al., 2013*). To elaborate, beyond the passive maintenance of incoming sensory information, WM provides an online mental workspace for active manipulation and control of stimulus contents (*Baddeley, 2003*; *Miller and Cohen, 2001*). As control functions often result in the generation and maintenance of new information, the brain needs to manage not only the original stimulus information but also the newly generated information in WM. Due to the limited cognitive resources available, it is likely that original stimulus representation in WM could adapt flexibly to various task goals beyond simple maintenance of WM contents, which might also co-vary with changes in the representation of the newly-generated information, leading to a systematic reconfiguration in representations of all levels across various cortices. We make two specific predictions from this account. First, in accordance with the findings of elevated neural activity in the frontal cortex with increasing demand for memory manipulation (*D'Esposito et al., 1999*, *D'Esposito et al., 2000*) and cognitive control (*Badre, 2008*; *Badre et al., 2010*; *Miller and Cohen, 2001*), stimulus representation in frontal cortex should be enhanced for active-control-related functions in WM. By contrast, due to the precise nature of stimulus representation in visual cortex, stimulus representation in this region should be enhanced for precise-maintenance-related functions in WM (*Henderson et al., 2022*; *Lee et al., 2013*). Second, within the brain regions that encode the newly generated information, a dynamic tradeoff between representations of original and new information should be observed to achieve a flexible allocation of limited cognitive resources (*Badre et al., 2021*; *Flesch et al., 2022*).

Using functional magnetic resonance imaging (fMRI), we directly tested this account by systematically investigating stimulus representation in visual, parietal, and frontal cortices during WM tasks with varying demands for active control. In particular, we surmised that stimulus representation in the frontal cortex would increase to accommodate complex control demands such as rule-based categorization. To this end, we employed a visual WM paradigm that required flexible switching between maintenance and categorization tasks. Specifically, in the maintenance task, participants maintained one visual orientation throughout a delay period, whereas in the categorization task participants were required to categorize the remembered orientation into one of two categories in accordance with previously learned rules, which could be either switched randomly between two rules on a block-by-block basis (Experiment 1) or fixed with one rule (Experiment 2). Thus, compared with the maintenance task, the categorization task imposed additional control demands of WM information at two different levels: the first level of control being stimulus categorization, because participants needed to adapt the stimulus based on the categorization rule in WM for subsequent category judgments; the second level of control being flexibility of rules, because with two categorization rules, the flexibility in control increased.

In line with our predictions, our results showed that stimulus information was more prominently represented in the frontal cortex during the categorization than the maintenance task, and this differential representation was enhanced with increasing demands for control. Importantly, the strength of stimulus representation in the frontal cortex was predictive of WM behavioral performance in the categorization but not in the maintenance task, implicating a selective involvement of the frontal cortex in control functions. By contrast, stimulus representation in the visual cortex was found to exhibit the opposite pattern, with higher strength for the maintenance than for the categorization task. Moreover, varying control demands across experiments revealed a dynamic tradeoff between stimulus and the newly-generated category representations. To further examine whether the enhanced stimulus representation in the frontal cortex during the categorization task could be explained by global coding strategy, we simulated our flexible WM tasks with multi-module recurrent neural networks (RNNs). The results of this computational modeling well replicated our human data when precise stimulus information was preserved at the output during network training. Taken together, our results indicate the importance of the frontal cortex for flexible control in WM and highlight the relative changes of stimulus representation in different cortical regions for varying task demands of WM.

## Results

### Behavioral learning and performance of WM tasks

In the fMRI session of Experiment 1, human participants (n=24) completed two tasks, maintenance and categorization, inside an MRI scanner. The maintenance task was a delayed match-to-sample WM task of orientations. Participants only needed to maintain the cued orientation throughout a memory delay. In the categorization task, participants also started with maintaining an orientation. After the task cue, they needed to categorize the remembered orientation into one of two categories using the cued categorization rule. Within an experimental block of nine trials, participants randomly switched between the two tasks. Across blocks, participants randomly switched between two categorization rules acquired during a preceding learning session. We randomized response mapping across trials to avoid potential influence by motor-planning signals (see *Figure 1A* for probe design). Prior to the main session, participants first completed a behavioral learning session to learn two categorization rules (Rule A and Rule B, see *Figure 1B*) that were orthogonal to each other. Participants acquired the two rules with equal familiarity ($t$(23) = 0.24, p=0.813; for Rule A, $M$=0.85, $SD$ = 0.050; for Rule B, $M$=0.85, $SD$ = 0.05; *Figure 1C*) and comparable precision (averaged error in reported boundaries for Rule A were 8.80°±6.29° and that for Rule B was 9.02°±5.69°; *Figure 1D*).

Overall, participants performed equally well on both tasks in the fMRI session. Accuracy for the maintenance task ($M ± SD$: 0.81±0.07) and that for the categorization task (0.82±0.05) did not significantly differ ($t$(23) = 1.51, p=0.144; *Figure 1E*), suggesting that the two tasks were matched in terms of task difficulty. In line with previous categorization studies demonstrating a boundary effect (*Ester et al., 2020*; *Freedman and Assad, 2006*), only in the categorization task but not in the maintenance task did participants perform better for trials distant from category boundaries in terms of both accuracy and reaction time (see *Figure 1G*). These results demonstrated the effect of categorization and confirmed that participants faithfully followed task instructions.

### Enhanced stimulus representation in frontal cortex during categorization task

The primary goal of this study was to determine the role of stimulus representation in various cortices in WM. Using conventional multivariate encoding and decoding methods, we tracked stimulus (i.e. orientation) representation in three brain regions of interest (ROIs) that have been implicated in representing WM information, including EVC, intraparietal sulcus (IPS), and superior precentral sulcus (sPCS) (*Christophel et al., 2018*; *Ester et al., 2015*; *Hallenbeck et al., 2021*; *Yu and Shim, 2017*), see *Figure 2—figure supplement 1* for a visualization of the anatomical locations of the ROIs.

First, we used multivariate inverted encoding models (IEMs) (*Brouwer and Heeger, 2009*; *Brouwer and Heeger, 2011*; *Ester et al., 2015*; *Rademaker et al., 2019*; *Yu and Shim, 2017*) to reconstruct orientation representation at the population level in each ROI. *Figure 2A* shows example orientation reconstructions from representative time points. To quantify the strength of orientation reconstructions, we calculated the reconstruction fidelity by first projecting the channel response at each

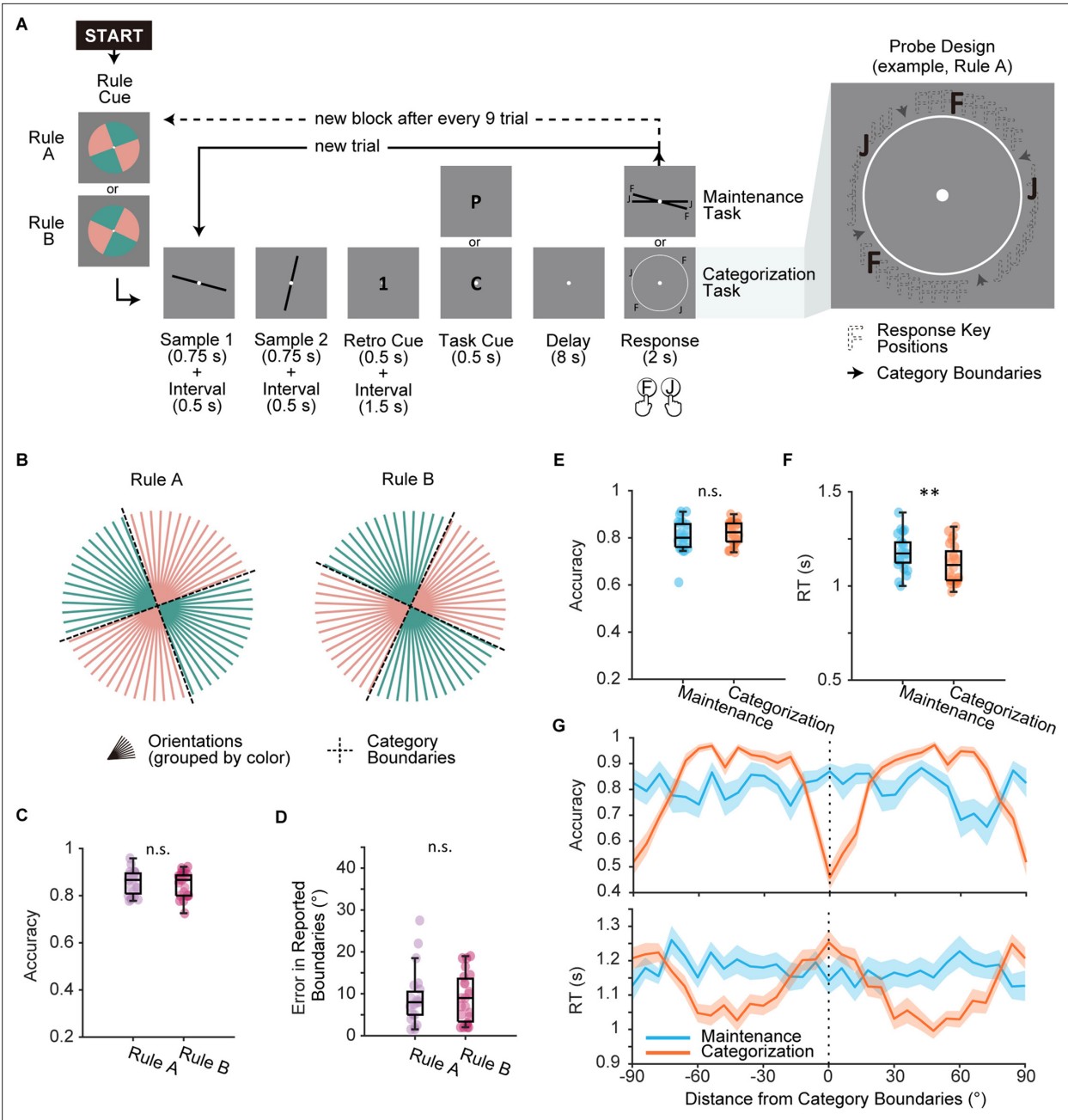

**Figure 1.** Experimental design and behavioral performance in Experiment 1. (**A**) Main task procedure. Each block started with a rule cue indicating the categorization rule for this block. On each trial, participants saw two orientations consecutively and were then cued to remember one of the orientations. In the maintenance task (cued by letter 'P'), participants needed to maintain the remembered orientation as precisely as possible. In the categorization task (cued by letter 'C'), participants needed to categorize the remembered orientation following the categorization rule of the current block. Maintenance and categorization trials were interleaved within an experimental block of nine trials. Categorization rule (Rule A or Rule B) switched randomly on a block-by-block basis. Response keys ('F' and 'J') for the categorization task were randomly assigned to the two categories. Each pair of keys displayed at random locations within the category to eliminate information on rule boundaries. (**B**) Illustration of the two orthogonal categorization rules (Rule A and Rule B). (**C**) Rule learning performance during learning session (n = 24) for Rule A (purple) and Rule B (pink). (**D**) Errors in participants' self-reported rule boundaries. Errors were calculated as the average distance from reported boundaries to ground truth boundaries. (**E**) Accuracy compared between tasks. Boxplots show the median and the 25th and 75th percentiles. Whiskers extend to 1.5 Inter quartile range (IQR) from the quartiles. Asterisks denote significant results, n.s.: not significant; **p<0.01. (**F**) Reaction time compared between tasks. Same conventions as (**E**). (**G**) Upper panel: accuracy in relation to distance from categorization boundaries. Lower panel: reaction time in relation to distance from categorization boundaries. Shaded areas represent ± SEM.

The online version of this article includes the following figure supplement(s) for figure 1:

**Figure supplement 1.** Behavioral performance of Experiment 2.

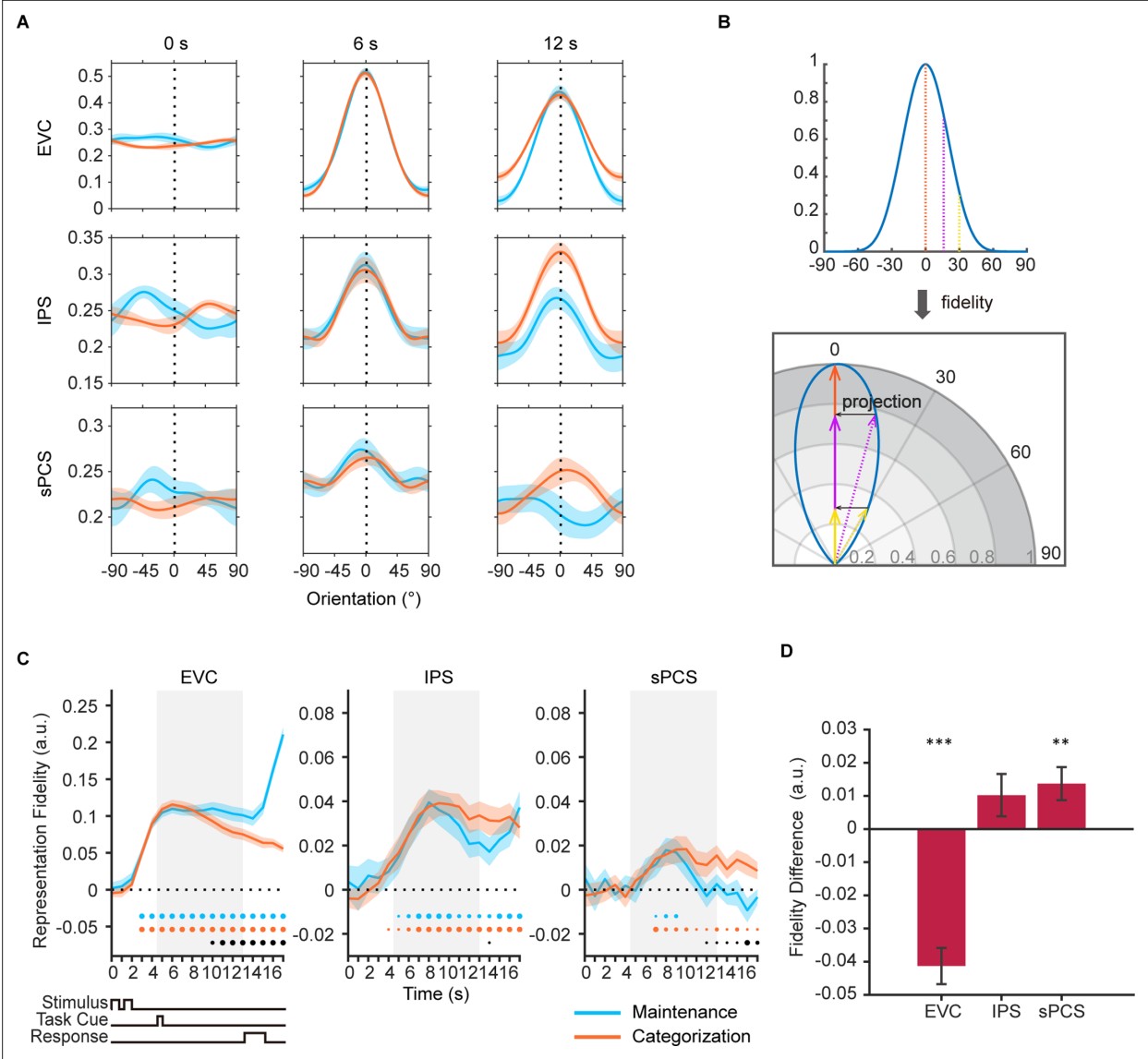

**Figure 2.** Orientation reconstructions at the population level using inverted encoding models (IEMs) in Experiment 1. (**A**) Reconstructed population-level orientation representations from selected time points in early visual cortex (EVC), intraparietal sulcus (IPS), and superior precentral sulcus (sPCS) for maintenance (blue) and categorization (orange) tasks, respectively (n = 24). X-axis represents distance from the cued orientation (at 0°), and y-axis represents reconstructed channel responses in arbitrary units. Significant orientation representation was observed at 6 s and 12 s but not at 0 s. Shaded areas represent ± SEM. (**B**) To quantify the strength of orientation reconstructions, we calculated the reconstruction fidelity by first projecting the channel response at each orientation onto a vector at the cued orientation and then averaging the projected vectors. (**C**) Time course of representational fidelity of orientations in EVC, IPS, and sPCS. In this and all subsequent figures with time course data, the time axis reflects raw time points without accounting for the hemodynamic response delay. Gray shaded areas indicate the entire memory delay following task cue. Blue and orange dots at the bottom indicate the FDR-corrected significance of representational fidelity at each time point of the corresponding task at p<0.05 (small), p<0.01 (medium), and p<0.001 (large). The bottom black dots indicate significant difference in representational fidelity between tasks (uncorrected). Horizontal dashed lines represent a baseline of 0. Shaded areas represent ± SEM. (**D**) Average difference of representational fidelity across 11–16 s in each region of interest (ROI) (from EVC to sPCS: p<0.00001, p=0.063, p=0.007, respectively). Positive difference indicates higher representational fidelity for the categorization task, and vice versa for negative difference. Black asterisks denote FDR-corrected significance, *p<0.05; **p<0.01; ***p<0.001. Error bars represent ± SEM.

The online version of this article includes the following source data and figure supplement(s) for figure 2:

**Source data 1.** p-values for the time course of inverted encoding model (IEM) results in *Figure 2*.

**Figure supplement 1.** Visualization of the anatomical locations of the regions of interest (ROIs) on the MNI brain.

**Figure supplement 2.** Control analyses for stimulus representation results in Experiment 1.

**Figure supplement 3.** Orientation reconstructions in primary motor cortex (M1) at the population level using IEMs.

orientation onto a vector at the cued orientation and then averaging the projected vectors to obtain the representational fidelity (*Rademaker et al., 2019*). A larger fidelity value indicates a stronger positive representation of orientation (See *Figure 2B* for illustration). *Figure 2C* demonstrates the time course of orientation reconstruction as quantified by representational fidelity. Critically, by comparing the representational fidelity values within the same ROI across conditions (maintenance vs. categorization), we minimized the impact of effect size on our comparisons between ROIs, as all the comparisons were primarily performed within the same ROI with a comparable effect size. In EVC, we found significant orientation representation in both maintenance and categorization tasks starting from the sample period (*Figure 2C* left panel; see *Figure 2—source data 1* for full statistics), even when the categorization task did not require explicit memory of stimulus information. Additionally, the strength of orientation representation in the maintenance task became significantly higher than that in the categorization task after the task cue during the delay, suggesting the strength of orientation representations in EVC reflected the degrees of task demands for maintaining visual details. In IPS, orientation representation was significant in both tasks, but did not differ from each other at most time points (*Figure 2C* middle panel). In sPCS, a reversed pattern was observed. In the maintenance task, orientation information was maintained during early delay period and then dropped to baseline level during late delay period. By contrast, in the categorization task, orientation representation was persistent throughout the delay and response periods. The strength of orientation representation in the categorization task became statistically higher than that in the maintenance task in late delay period (*Figure 2C* right panel), suggesting that this differential representations of visual stimulus in the frontal cortex reflected the demand for active control of memory contents. To facilitate comparison of the differential stimulus representation across ROIs, we averaged the difference in representational strength across a late task epoch (11–16 s), and the difference in stimulus representation between ROIs remained (*Figure 2D*).

We validated the difference in stimulus representations through a series of control analyses. First, we demonstrated that these results cannot be explained by the specific model used to train the data (*Liu et al., 2018*; *Sprague et al., 2018*) nor the specific analytical approach used, because similar patterns were observed when we trained the IEM separately for each condition (*Figure 2—figure supplement 2A*) or adopted a Support Vector Machine (SVM) decoding approach (*Figure 2—figure supplement 2B*; *Henderson et al., 2022*; *Rademaker et al., 2019*). Mean activation differences between tasks cannot account for the results either, because when we removed the mean differences in BOLD activity between tasks, the difference in representational strength remained (*Figure 2—figure supplement 2C*). Furthermore, to remove the potential impact of voxel number on IEM, we selected the top 500 of most sample- or delay-selective voxels from each ROI and trained IEM using the selected voxels. Again, this analysis yielded similar findings (*Figure 2—figure supplement 2D*). Lastly, when repeating the IEM analyses in primary motor cortex (M1), no similar patterns were observed, suggesting that our results cannot be explained by motor- or response-related activity (*Figure 2—figure supplement 3*).

Together, these results demonstrated enhanced stimulus representation in the frontal cortex with increased demand for active control, as well as those in the visual cortex with increased demand for precise WM maintenance.

## Prediction of categorization behavior by frontal stimulus representation

Previous WM studies have shown that the strength of stimulus representation in EVC positively correlated with memory performance (*Emrich et al., 2013*; *Ester et al., 2013*; *Gosseries et al., 2018*), suggesting that EVC plays an important role in precise WM maintenance. However, stimulus representation in the frontal cortex rarely predicted behavioral performance in maintenance task (*Hallenbeck et al., 2021*), although univariate activation in the same brain region can predict memory-guided saccade performance (*Curtis et al., 2004*). Nevertheless, if frontal stimulus representation is involved in WM control, its behavioral relevance should be subject to observation with increased control demands. Therefore, we assessed the behavioral predictability of stimulus representation during the delay period in EVC, IPS, and sPCS. *Figure 3A, B and C* illustrate the time course of correlation results, results from representative time points, and results collapsed across the late epoch, respectively. Consistent with previous findings, we found the strength of stimulus representation in

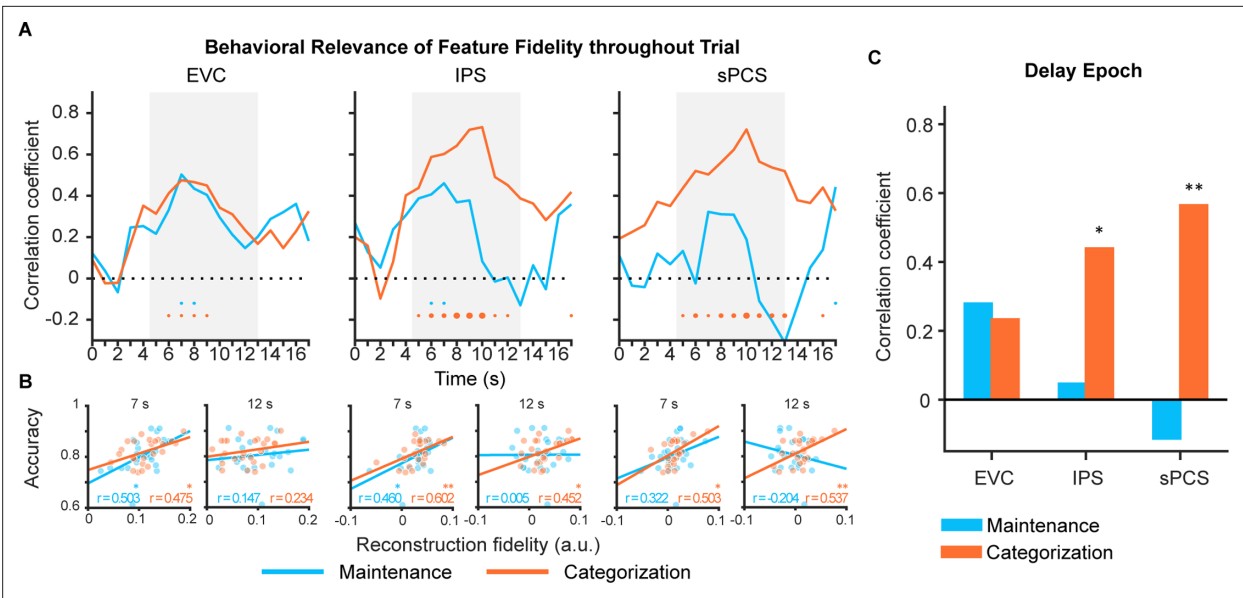

**Figure 3.** Behavioral correlation of stimulus representation for maintenance (blue) and categorization (orange) tasks in Experiment 1. (**A**) Time course of correlation coefficients between behavioral performance and orientation representational fidelity (n = 24) in early visual cortex (EVC), intraparietal sulcus (IPS), and superior precentral sulcus (sPCS). Gray shaded areas indicate the entire memory delay following task cue. Blue and orange dots at the bottom indicate significance of correlation (uncorrected) at each time point at p<0.05 (small), p<0.01 (medium), and p<0.001 (large). (**B**) Correlation scatter plots at representative time points (7 s and at 12 s) in EVC, IPS, and sPCS. R denotes Pearson correlation coefficients. (**C**) Correlation between behavioral performance and orientation representational fidelity collapsed across the late epoch (11–16 s). Asterisks denote significant results, *p<0.05; **p<0.01.

The online version of this article includes the following source data and figure supplement(s) for figure 3:

**Source data 1.** p-values for the time course of correlation results in *Figure 3*.

**Figure supplement 1.** Behavioral correlation of stimulus representation in Experiment 2.

EVC during the early delay period predicted behavioral accuracies in both maintenance and categorization tasks (see *Figure 3—source data 1* for full statistics). Similar predictability was found in IPS, with stimulus representation predicted behavior in the maintenance task during early delay and in the categorization task throughout the entire memory delay. Interestingly, we found that, throughout the entire memory delay, the strength of stimulus representation in sPCS predicted behavioral accuracies only in the categorization task but not in the maintenance task. These results highlighted the functional significance of stimulus representation in sPCS exclusively for the categorization task.

## Reduced frontal stimulus representation with lower control demand

In Experiment 1, participants flexibly switched between two categorization rules to prompt the manipulation of WM content on a trial-by-trial basis. The rule switching increased control demand but also complicated the interpretation of our results. To exclude potential impact of rule switching, we conducted Experiment 2 (n = 24), in which participants performed the maintenance and categorization tasks with only one fixed rule. Behavioral results of Experiment 2 again demonstrated a classic boundary effect and were comparable to Experiment 1, with no significant difference between experiments in terms of either accuracy or reaction time (*Fs* <1.28, *ps* >0.26; *Figure 1—figure supplement 1*). When using IEMs to reconstruct stimulus representation, we found EVC and IPS both showed patterns similar to those in Experiment 1 (*Figure 4A* left and middle panels), with stimulus representation decreased in EVC in the categorization task and remained at the same level in IPS between the two tasks (see *Figure 4—source data 1* for full statistics). The frontal region, sPCS, also showed a differential enhancement of stimulus representation in the categorization task as in Experiment 1, but in an earlier period (*Figure 4A* right panel). To validate such a temporal difference, we defined an additional early task epoch (5–10 s), and confirmed a significant difference in stimulus representation in sPCS during the early (p=0.015; *Figure 4B*) but not during the late epoch (p=0.372; *Figure 4C*). In addition, we performed a mixed ANOVA on experiments (Experiment 1 vs 2) and epochs (early vs. late epoch) and observed a significant interaction effect between the two, $F(1, 46)=7.43$, p=0.009,

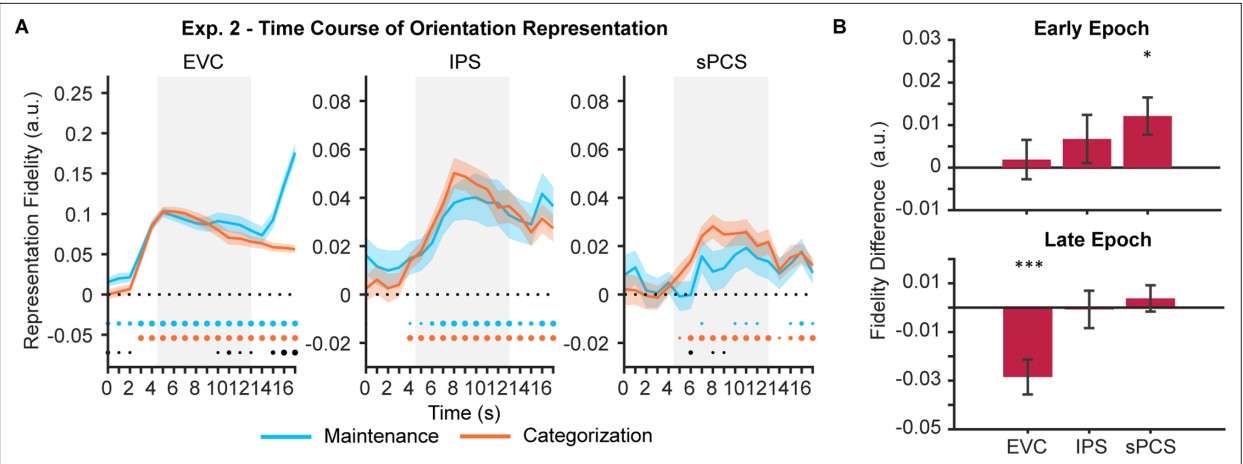

**Figure 4.** Orientation reconstructions at the population level using inverted encoding models (IEMs) in Experiment 2. (**A**) Time course of representational fidelity of orientations (n = 24) in early visual cortex (EVC), intraparietal sulcus (IPS), and superior precentral sulcus (sPCS). Gray shaded areas indicate the entire memory delay following task cue. Blue and orange dots at the bottom indicate the FDR-corrected significance of representational fidelity at each time point of the corresponding task at p<0.05 (small), p<0.01 (medium), and p<0.001 (large). The bottom black dots indicate significant difference in representational fidelity between tasks (uncorrected). Horizontal dashed lines represent a baseline of 0. Shaded areas represent ± SEM. (**B**) Average difference of representational fidelity across an early (top, 5–10 s) and a late (bottom, 11–16 s) task epoch in each region of interest (ROI). Positive difference indicates higher representational fidelity for the categorization task, and vice versa for negative difference (FDR-corrected). Error bars represent ± SEM. *p<0.05; ***p<0.001.

The online version of this article includes the following source data for figure 4:

**Source data 1.** p-values for the time course of inverted encoding model (IEM) results in *Figure 4*.

suggesting that the two experiments differed in terms of the temporal emergence of the differential stimulus representation in the frontal cortex. Taken together, these results are consistent with our expectation that, with reduced control demand, the differential enhancement of stimulus representation in frontal cortex was still present but decreased during late memory delay. Nevertheless, stimulus representation in Experiment 2 still predicted behavioral performance as in Experiment 1, although the difference between tasks was reduced (*Figure 3—figure supplement 1*).

## Category representation in WM in various cortices

Having observed a differential representation of stimulus in the frontal cortex, we next asked how newly generated information in WM during the categorization task emerged and sustained in the distributed WM network and how representations of the original stimulus and the new information interacted. The categorization task could demand additional generation of category information in WM. We, therefore, trained SVMs to decode category information during the categorization task. For each rule, the SVM decoder was trained to discriminate between the two categories. In both experiments, we found that during the late epoch, category information could be well decoded across ROIs (ps <0.044, *Figure 5A*; also see *Figure 5—figure supplement 1* for full decoding time course), with a marginal difference between experiments in sPCS (p=0.055).

One might argue that the category decoding results could at least be partially attributed to stimulus similarity. To minimize the impact of stimulus similarity on category decoding, we additionally trained another decoder using the opposite rule (i.e. using category labels from the orthogonal rule; see *Figure 5B* for results). We then calculated an abstract category index by subtracting decoding accuracy under the opposite rule from that under the true rule (*Mok and Love, 2020*; *Figure 5C*). The rationale was that the amount of stimulus similarity would be comparable for the opposite rule, but additional category information, if existed, should result in higher decoding accuracy for the true rule. In other words, positive abstract category index indicates evidence for stimulus-independent category representation. After removing stimulus-related signals, average decoding performance of abstract category was only evident in Experiment 2 (*ps* <0.017) but not in Experiment 1 (*ps* >0.14) for all ROIs. Moreover, decoding performance of abstract category was significantly higher in Experiment 2 than Experiment 1 in sPCS (*P*=0.034; *Figure 5D*). These results together suggest a potential

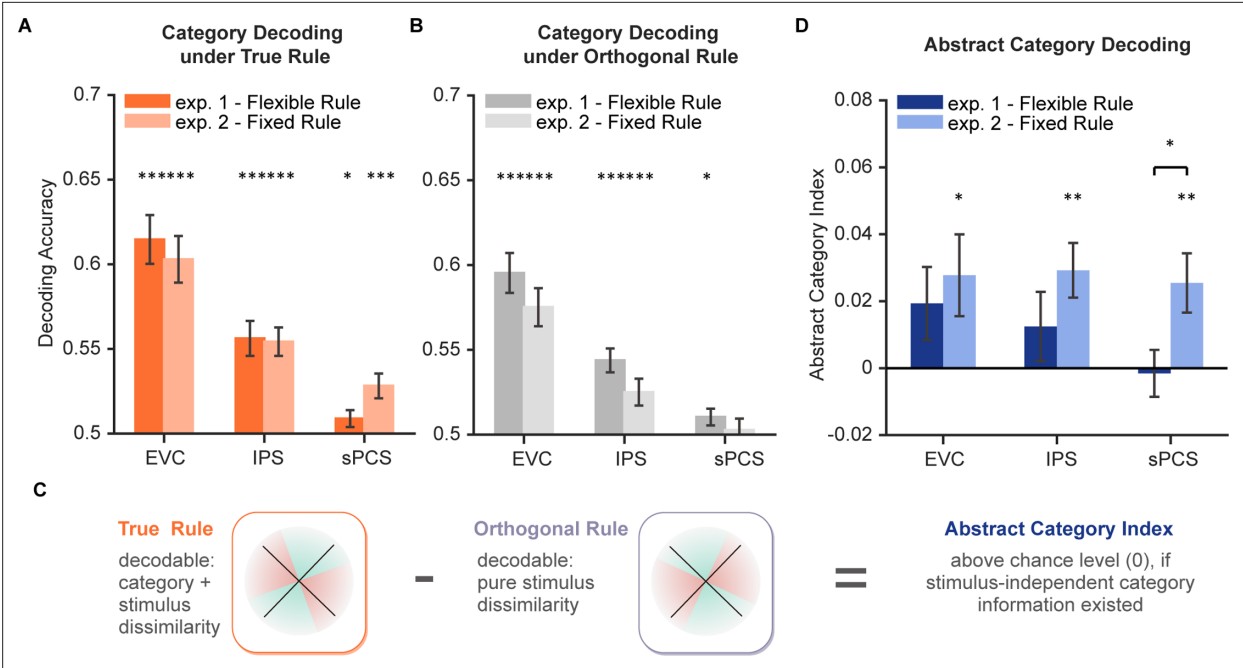

**Figure 5.** Decoding performance for category and abstract category information. (**A**) Average category decoding accuracy using category labels under true rule across the late task epoch (11–16 s) in each region of interest (ROI) of both experiments (n = 24 for each experiment). (**B**) Average category decoding accuracy using category labels under orthogonal rule across the late task epoch (11–16 s) in each ROI of both experiments. (**C**) Schematic illustration of abstract category decoding. In the categorization task, category information can be decoded using category labels according to the true categorization rule. On the other hand, category can also be decoded due to stimulus similarity. Thus, to remove stimulus-dependent categorical information, we calculated an abstract category index by removing decoding accuracy using orthogonal category boundaries (assuming comparable stimulus-dependent effect) from that using true rule boundaries. (**D**) Average abstract category decoding index across the late task epoch (11–16 s) in each ROI of both experiments. Black asterisks denote FDR-corrected significance, *p<0.05; **p<0.01; ***p<0.001. Error bars represent ± SEM.

The online version of this article includes the following figure supplement(s) for figure 5:

**Figure supplement 1.** Time course of category and abstract category decoding performance in Experiment 1 and 2.

**Figure supplement 2.** Delineating stimulus and category effects using linear mixed-effects modeling.

**Figure supplement 3.** Stimulus, category, and abstract category results in additional frontal regions in Experiment 1 and 2.

tradeoff between stimulus difference and category representation in the frontal cortex. This tradeoff was further depicted using linear mixed-effects modeling (LMEM) on representational dissimilarity matrices (RDMs) of neural activities to decouple the contributions of stimulus and category to neural representation (*Figure 5—figure supplement 2A*). Overall, the LMEM results (*Figure 5—figure supplement 2B–D*) replicated the above findings, with significant stimulus but not category representation in sPCS in Experiment 1, and a decreased contribution of stimulus but emerging category representation in the same brain region in Experiment 2.

Lastly, for completeness, we repeated the IEM and category decoding analyses on additional ROIs in the frontal cortex, to investigate whether the observed results were specific to sPCS. Specifically, we defined three additional ROIs based on the HCP atlas (*Glasser et al., 2016*), including the inferior precentral sulcus (iPCS), inferior frontal sulcus (IFS), and middle frontal gyrus (MFG) (*Ester et al., 2015*; *Mackey et al., 2017*; *Yu and Shim, 2017*). Overall, results in the three ROIs were comparably weaker than sPCS (*Figure 5—figure supplement 3*). There was some indication that the MFG might share some results for orientation representation and category decoding, although this pattern was weaker and was only observed in some analyses in Experiment 2.

## Differential stimulus representation in frontal cortex replicated by RNN modeling

Finally, we tested how stimulus representation could emerge in frontal cortex at the mechanistic level using RNN models. Our hypothesis is that precise stimulus representation during WM might

emerge in frontal cortex in response to complex task demands such as rule-based categorization. In other words, instead of relying (solely) on category representations, the cortical network might have adopted a different strategy to accommodate flexible task requirements in the current study, for instance, by preserving stimulus information until a later stage of information processing. This different strategy can be implemented by altering the RNN's output structure. Therefore, the logic of this modeling analysis was to examine whether explicitly placing a demand for the model to preserve stimulus representation would recapitulate our fMRI findings in frontal cortex, in comparison to a model that did not specify such a demand.

Two types of modular RNNs were trained on the maintenance and categorization tasks simultaneously (*Masse et al., 2019*; *Zhou et al., 2021*). The networks shared common input and hidden layer structures (i.e. orientation-tuned and retro/task cue-related units as the input layer, recurrent units with short-term synaptic plasticity in the hidden layer [80% excitatory +20% inhibitory units, equally distributed in three separate modules]). The only difference was in the structure of the output layer. The first type of RNN (RNN1; n=20) had only two units in the output layer to indicate networks' choice (*Figure 6A*), whereas the second type of RNN (RNN2; n=20) had additional units in the output layer corresponding to the original stimulus information. In other words, the second RNN was designed to maintain stimulus information throughout the network modules. For the common hidden layer, we included three hierarchically organized (posterior, middle, and anterior) modules of recurrent units generated according to neurobiological principles of neuronal connections (e.g. denser connectivity within than between modules) to simulate the interconnected brain areas in our ROI-based fMRI analyses above: the posterior module (Module 1, simulating EVC) was directly connected with the input layer, the middle module (Module 2, simulating IPS) received projections from the posterior module and relaying information to the anterior module, and the anterior module (Module 3, simulating sPCS) projected to the output layer. Task events were simulated as numerical inputs to the model, matching the procedures of Experiment 1 (see Methods for details).

After successful training, defined as reaching at least 90% accuracy in all tasks in the same training batch, we applied an SVM decoding approach to investigate population-level stimulus representations in neuronal spiking activities of the RNNs. We found that in RNN1, both the middle and anterior modules showed stronger stimulus representation in the maintenance task than the categorization task during the delay period ($p_{posterior}$ = 0.09, $p_{middle}$ = 0.011, $p_{anterior}$ = 0.007; *Figure 6B* upper panel), opposite to our fMRI observation in IPS and sPCS (*Figure 6B* inset). In comparison, decoding performance in RNN2, which was explicitly required to maintain stimulus information for the output, yielded results consistent with our human findings, with increased stimulus decoding performance during categorization only in the anterior module ($p_{posterior}$ = 0.436, $p_{middle}$ = 0.212, $p_{anterior}$ = 0.026; *Figure 6B* lower panel).

Besides the difference in stimulus representation, we further tested whether RNN2 could also replicate the human results on category representation. For this analysis, we focused on abstract category representation to fully remove the impact of stimulus on category decoding. To examine the influence of control demand on category decoding, following our fMRI experiment, we trained 20 additional RNNs with the same output structure as RNN2 (preserving stimulus information) to perform the tasks with a fixed categorization rule, mimicking the task structure of Experiment 2. Consistent with our human findings, we observed increased abstract category decoding performance in the fixed-rule models compared to the flexible-rule RNNs, throughout the modules ($p_{posterior}$ = 0.045, $p_{middle}$ = 0.003, $p_{anterior}$ <0.001; *Figure 6C* lower panel). Similar differences between fixed-rule and flexible-rule models were also observed in RNN1 (all *p*s <0.001; *Figure 6C* upper panel).

To quantify the similarity between human and RNN in terms of stimulus and category representations, we performed representational similarity analysis (RSA) comparing human and RNN data across tasks. Specifically, we aligned the RDMs by either stimulus (*Figure 7A*) or category (*Figure 7C*) to prioritize the comparison of stimulus or category representations, respectively. Overall, across experiments, RNN2 consistently demonstrated higher similarity to human data than RNN1, particularly in Module 3 that corresponds to the frontal cortex (*Figure 7B and D*). In terms of the RDM patterns, consistent with previous results, RNN2 exhibited stronger stimulus representation than RNN1. Moreover, fixed-rule RNNs were found to be more categorical than flexible-rule RNNs regardless of the output structure. Altogether, these findings demonstrated that our fMRI results could be simulated by RNN models when stimulus information for readout was preserved, suggesting that the requirement

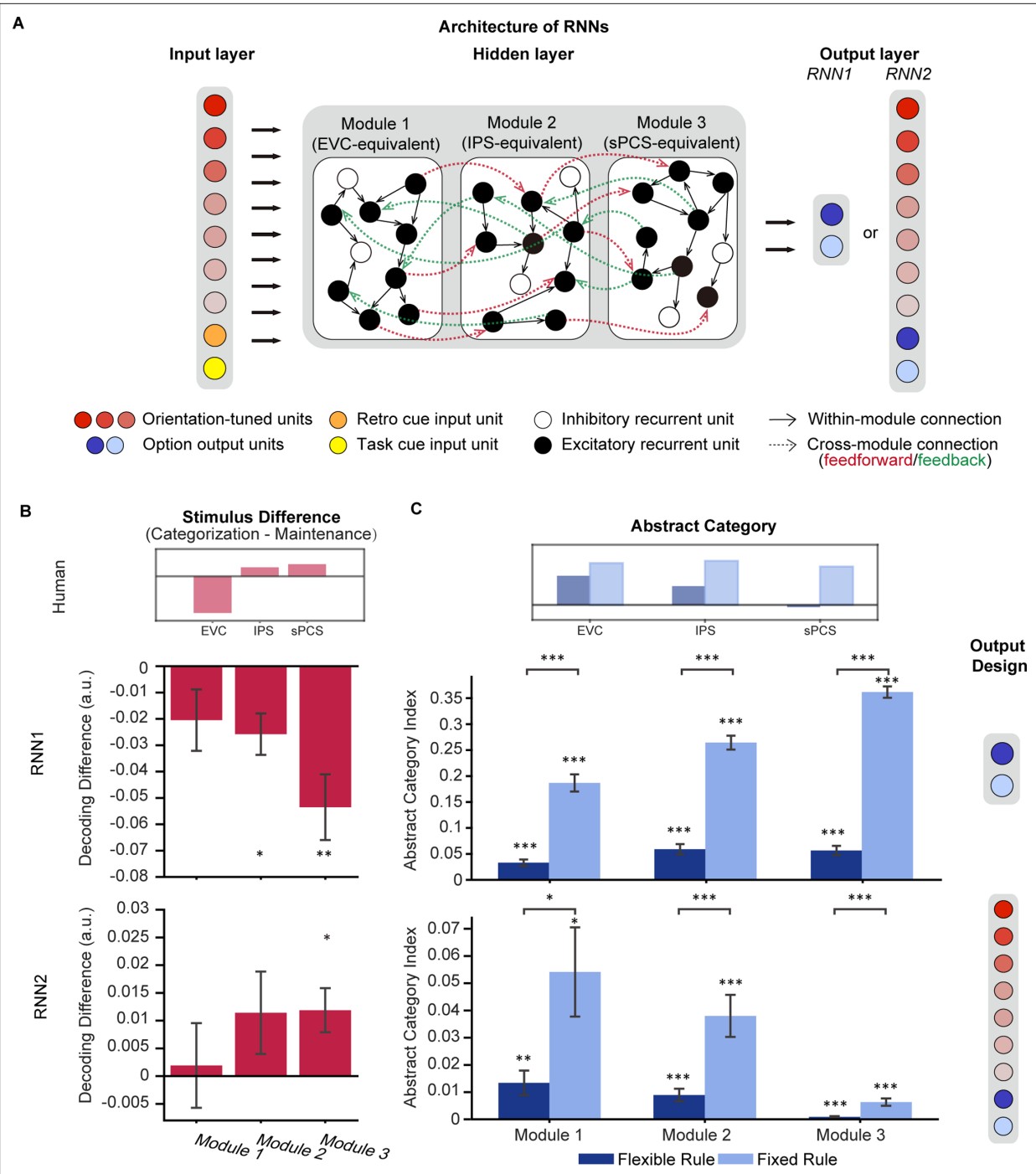

**Figure 6.** Architecture of recurrent neural networks (RNNs) and simulation results. (**A**) All networks consist of three layers of artificial units: the input, hidden, and output layers. For both RNN1 and RNN2, the input layer contains 20 units including 15 orientation-tuned, red units and 5 cue units (retrocue and task cue, orange and yellow). The hidden layer consists of three modules of 200 recurrent units with short-term synaptic plasticity (STSP), further divided into 80% excitatory (black) and 20% inhibitory (white). Connectivity within each module (black arrow) is denser compared to between modules (red and green arrows), which only occur between excitatory units. Only excitatory units in module 1 receive projections from the input layer and only excitatory units in module 3 project to the output units. For RNN1, networks output (0,1) or (1,0) through the 2 units in the output layer to indicate responses. For RNN2, the network output (0,1) or (1,0) to report the category to which the cued orientation belonged in the categorization task, or (0,0) in the maintenance task (blue units). Importantly, the models also output the orientation itself through 15 additional orientation-tuned units (red). (**B**) Difference in stimulus decoding between tasks in RNN1 (upper panel; n = 20) and RNN2 (lower panel; n = 20). Results were averaged across the delay period. Positive difference indicates higher decoding accuracy for the categorization task, and negative difference indicates higher decoding accuracy for maintenance. The inset above illustrates stimulus difference in human fMRI results during late epoch in Experiment 1 to provide a reference

*Figure 6 continued on next page*

*Figure 6 continued*

for expected patterns in RNNs. (**C**) Average abstract category information across the delay period for RNN1 (upper panel) and RNN2 (lower panel). The inset above illustrates abstract category representation in human fMRI. Error bars represent ± SEM. Black asterisks denote FDR-corrected significance, *p<0.05; **p<0.01; ***p<0.001.

The online version of this article includes the following figure supplement(s) for figure 6:

**Figure supplement 1.** Recurrent neural network (RNN) results using inverted encoding models (IEMs).

for flexible control of WM content could demand high-fidelity stimulus representation at the output stage of the model. Notably, we found that RNN2 generally took fewer iterations for training and had fewer failures in learning the task (with a defined maximal number of iterations).

## Discussion

In this study, we investigated the emergence and maintenance of stimulus representation with varied control demands of WM. In a distributed human cortical network encompassing visual, parietal, and frontal cortex, we found enhanced stimulus representation in the frontal cortex that tracked increasing demands on active WM control, as well as enhanced stimulus representation in the visual cortex that tracked the demand for the precise maintenance of WM content. The enhanced stimulus representation in frontal cortex was well predicted by RNNs that preserved stimulus information for readout at the output stage. Together, these results highlight the unique and critical contributions of stimulus representations in different cortical regions for distinct aspects of WM, and help to resolve the current controversy in the roles of various cortices in WM (see *Figure 8* for a schematic summary of the fMRI findings).

### Role of visual cortex in WM maintenance

The visual cortex has been considered a critical site for maintaining visual WM in the context of sensorimotor recruitment hypothesis (*D'Esposito and Postle, 2015*; *Harrison and Tong, 2009*). This idea, however, has been challenged in recent years due to some seemingly contradictory findings from the human neuroimaging studies. For example, compared to the frontoparietal cortex, mnemonic representations in EVC were found to be more vulnerable to distractors (*Bettencourt and Xu, 2016*; *Hallenbeck et al., 2021*; *Lorenc et al., 2018*). The decodability of memory contents in visual cortex also depends on the specific task type. A previous study showed that in nonvisual tasks that required judgments on object category instead of visual details, memory contents were no longer decodable in the visual cortex (*Lee et al., 2013*). In this study, we found that, although the strength of stimulus representation in EVC differed between WM maintenance and categorization tasks, a copy of stimulus representation remained in EVC during the categorization task. Moreover, stimulus representations in both tasks were equally predictive of subsequent memory performance, suggesting the functional significance of EVC representations in WM.

The discrepancy between our results and that of the previous work (*Lee et al., 2013*) could be attributed to the fact that our categorization task required participants to manipulate remembered information according to arbitrary yet flexible categorization rules, rather than simply paying selective attention to different aspects (visual details vs. category membership) of everyday objects. In our case, maintaining visual details of the memoranda was critical for accurate behavioral responses. Our finding is consistent with the prediction of sensorimotor recruitment hypothesis that representation of memory contents in the visual cortex is necessary for the precise maintenance of visual information. The observation of robust category representation in EVC during the response period further indicated the recruitment of EVC in categorization, possibly for boundary comparison and rule implementation. In fact, our results are consistent with a recent study demonstrating significant stimulus representation in EVC even when memoranda had been transformed into a motor format (*Henderson et al., 2022*). In addition, electrophysiological research in non-human primates has also shown robust feature selectivity in the visual cortex during a categorization task (*Brincat et al., 2018*), and recent computational modeling work has suggested intact maintenance of sensory information during categorical judgments (*Luu and Stocker, 2021*).

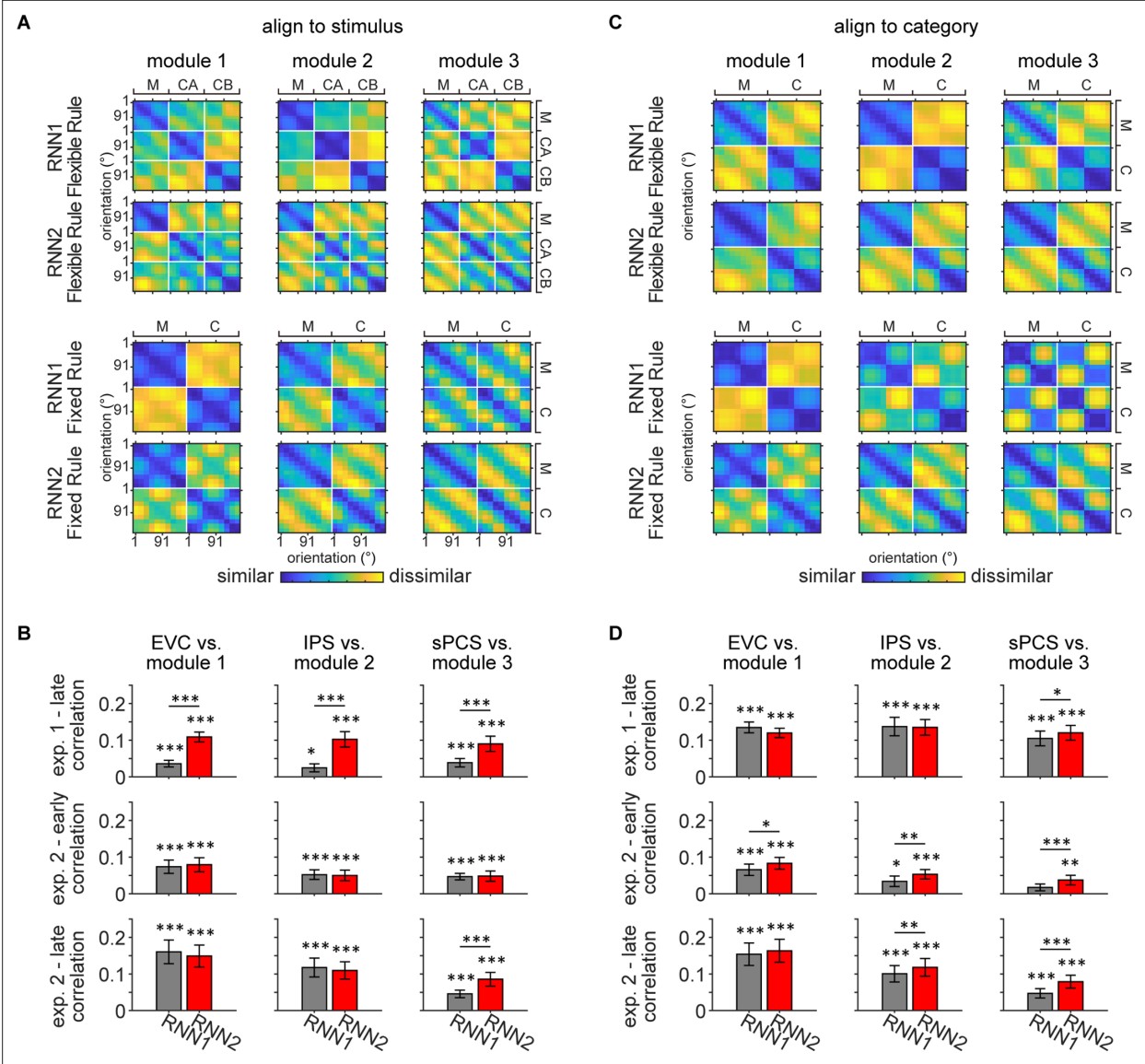

**Figure 7.** Representational similarity between recurrent neural network (RNN) and human data. (**A**) Averaged representational dissimilarity matrices (RDMs) for RNN1 and RNN2 under flexible rule (upper panel; n = 20) and fixed rule (lower panel; n = 20), with data aligned to stimulus. Individual RDM was constructed by calculating the Pearson correlation distance between activation across trials and averaging across the delay period for each orientation and task. Before averaging, each RDM was aligned to stimulus, with orientations sorted from 1° to 180° and ordered by tasks (maintenance [M], categorization-Rule A [CA], and categorization-Rule B [CB] for flexible rule; maintenance and categorization [C] for fixed rule). (**B**) To quantify the similarity between RNN and human data, Kendall's Tau correlation coefficients were computed between the RDMs of RNN and human data. Comparisons were performed between flexible-rule RNN and human data from Experiment 1 (top row), and between fixed-rule RNN and human data from early and late epochs of Experiment 2 (middle and bottom rows), for each module and corresponding ROI. Significance of correlation was evaluated using one-sided signed-rank tests. Difference between RNN1 and RNN2 was calculated using Z-transformed correlation coefficient and permutation tests. Black asterisks denote uncorrected significance, *p<0.05; **p<0.01; ***p<0.001. Error bars represent ± SEM. (**C**) Same as (**A**) but with data aligned to category. RDMs for each task were aligned to category and rules, with the first half of rows and columns corresponding to one category and the other half to the other category. RDMs for the two categorization rules were further averaged, resulting in two tasks (M and C). (**D**) Same conventions as (**B**).

## Role of frontal cortex in active WM control

Compared to the prominent role of EVC in memory maintenance, sPCS in the frontal cortex played a dominant role in WM tasks that require active control of memory contents such as categorization. Although stimulus representations in sPCS have been observed during WM in previous studies, the nature of these representations remained debatable. In WM tasks that required mere maintenance of

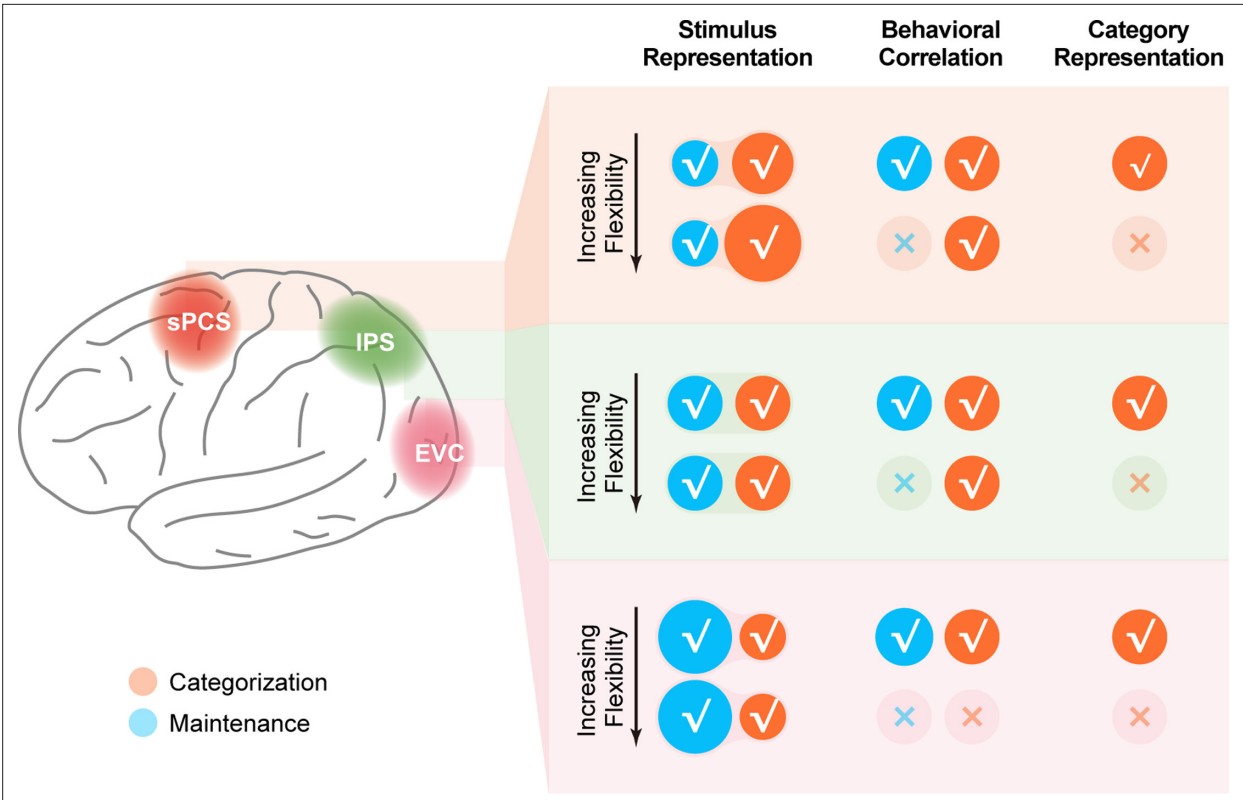

**Figure 8.** Schematic summary of the involvement of different cortical regions in the current study. Schematic summary of stimulus representation and corresponding behavioral correlation, as well as category representation in early visual cortex (EVC), intraparietal sulcus (IPS), and superior precentral sulcus (sPCS) for both Experiment 1 and Experiment 2. √ denotes the presence of the corresponding finding, and × denotes the absence of the corresponding finding in maintenance (blue) and categorization (orange) tasks.

memoranda, stimulus was not always decodable in the frontal cortex (*Emrich et al., 2013*; *Gosseries et al., 2018*; *Riggall and Postle, 2012*), raising the issue of functional significance of stimulus representation in the frontal cortex. On the other hand, stimulus representation in the frontal cortex could become robust in the face of tasks that require attentional prioritization and extensive training (*Bettencourt and Xu, 2016*; *Christophel et al., 2018*; *Hallenbeck et al., 2021*; *Lorenc et al., 2018*; *Miller et al., 2022*). Our current study contributes to the resolution of this issue by demonstrating that stimulus representation in sPCS increased with increasing demands for WM control. This finding is in line with recent computational studies proposing that active WM functions may involve neuronal mechanisms different from that for passive maintenance. For example, passive maintenance could rely mainly on synaptic plasticity mechanisms, whereas active control functions such as distractor resistance and information manipulation involve more neuronal spiking activity (*Masse et al., 2019*; *Wang, 2021*). In this study, we provided the first empirical evidence that the frontal cortex exhibits enhanced stimulus representation in categorization task requiring active WM control and this representation is predictive of WM performance. In contrast, stimulus representation of WM maintenance failed to predict WM performance at high control demand. Moreover, by examining additional ROIs in the frontal cortex (i.e. iPCS, IFS, and MFG), we found that results in these brain regions were overall weaker, with the MFG demonstrating the most comparable, albeit much weaker, results to sPCS. The specific involvement of sPCS in our experiment could be due to the type of stimulus (i.e. orientation) we used, as previous work has highlighted a prominent role of sPCS in encoding stimulus-specific information in spatial (*Hallenbeck et al., 2021*; *Sprague and Serences, 2013*) and orientation WM (*Ester et al., 2015*; *Yu and Shim, 2017*; *Yu and Shim, 2019*), but to a lesser extent in other WM tasks such as color (*Yu and Shim, 2017*; *Yu and Shim, 2019*). It would be of interest to further investigate whether this active control in the sPCS could be generalized to other frontal regions and tasks that

require other types of WM control such as mental rotation (*Shi and Yu, 2024*), and how other types of control task may adapt to changing flexible control demands.

## WM representations in frontal cortex support cognitive flexibility

Our results in the frontal cortex are also in line with recent theoretical proposals in the field of cognitive flexibility. To behave flexibly in complex environments with limited cognitive resources, two mechanisms have been proposed: low-dimensional abstraction of stimulus representation for generalization and efficient learning, and high-dimensional stimulus representation for separability and flexible readout (*Badre et al., 2021*; *Flesch et al., 2022*; *Fusi et al., 2016*). Within this framework, high-dimensional stimulus representation during WM might emerge in the frontal cortex in response to complex control demands such as rule-based categorization. The results of the two fMRI experiments in the current study jointly demonstrate a dynamic tradeoff between high-dimensional stimulus and low-dimensional category representations depending on the control demand. Specifically, when control demand was reduced with a single categorization rule in Experiment 2 compared to Experiment 1, the differential stimulus representation in the frontal cortex was also reduced during the late delay period, accompanied by an increase in category decoding performance especially in the frontal cortex. This result is consistent with neurophysiological findings in non-human primates: while robust category selectivity was observed in frontoparietal cortex during the delay period of categorization tasks when the animal was trained on the categorization task only (*Brincat et al., 2018*; *Freedman and Assad, 2006*; *Freedman et al., 2001*; *McKee et al., 2014*), category selectivity in the parietal cortex was significantly reduced when the animal had been exposed to a maintenance task prior to categorization training (*Latimer and Freedman, 2023*). Our RNN simulation further confirmed that this dynamic reconfiguration in information coding at the network level can be well explained by a change in the coding strategy for the network readout. In other words, in flexible environments, and with rich prior experience, the brain might adopt an entirely different strategy for processing information in WM. High-dimensional stimulus information might be preserved in its original identity in the higher-order cortex, potentially reducing processing demands in dealing with each task and thereby facilitating efficiency and flexibility (*Badre et al., 2021*; *Flesch et al., 2022*; *Fusi et al., 2016*). One important future direction would be to further address the meta-control mechanisms that determine the flexible selection of coding strategies for WM (*Eppinger et al., 2021*).

## Differentiating between frontal and parietal cortex in WM functions

While many previous WM studies have focused on the functional distinction between sensory and frontoparietal cortex, it has remained less clear how frontal and parietal cortices might differ in terms of WM functions. Some studies have reported stimulus representations with similar functionality in frontal and parietal cortex (*Christophel et al., 2018*; *Yu and Shim, 2019*), while others have observed differential patterns (*Hu and Yu, 2023*; *Lee et al., 2013*; *Li et al., 2023*). We interpret the differential patterns as reflecting a difference in the potential origin of the corresponding cognitive functions. For example, in our study, sPCS demonstrated the most prominent effect for enhanced stimulus representation during categorization as well as the tradeoff between stimulus difference and category representation, suggesting that sPCS might serve as the source region for such effects. On the other hand, IPS did show visually similar patterns to sPCS in some analyses. For instance, stimulus representation in IPS was visually but not statistically higher in the categorization task. Additionally, stimulus representation in IPS also predicted behavioral performance in the categorization task. These results together support the view that our findings in sPCS do not occur in isolation, but rather reflect a dynamic reconfiguration of functional gradients along the cortical hierarchy from early visual to parietal and then to frontal cortex.

## The alignment between RNN and fMRI results

Although the current RNNs effectively captured our key fMRI findings, including increased stimulus representation in the frontal cortex as well as the tradeoff in category representation with varying control demands, we acknowledge that differences remain between the two modalities. For instance, all three RNN modules demonstrated significant abstract category decoding as well as differences in category decoding between experiments, which differed from our fMRI results. This discrepancy could partially be due to the higher signal sensitivity in RNNs. In addition, RNN2 did not show decreased

stimulus decoding for categorization in the EVC module. However, we found that applying IEMs to the RNN data revealed a similar negative trend in the EVC module (*Figure 6—figure supplement 1*), though it was not statistically significant. This result suggests that any negative difference between categorization and maintenance in EVC module was weaker compared to fMRI, if existed. We speculate that enhancing the negative difference in the EVC module might require additional modules or inputs to strengthen fine-grained stimulus representation in EVC, a mechanism that could be of interest for future research.

## Conclusion

In conclusion, we observed a distributed cortical network, including early visual, parietal, and frontal cortex, in representing stimulus-specific information in WM. These stimulus representations in visual and frontal cortex played distinct functional roles, with those in EVC contributing primarily to precise maintenance and those in frontal cortex contributing primarily to active control in WM. RNN simulations indicated that the stimulus representation in the frontal cortex might have emerged as a result of output selection to facilitate cognitive flexibility. Collectively, these results help to reconcile current debates on the functional roles of different cortical regions in WM, and provide new insights into how a unified WM framework could support varied control demands.

# Materials and methods

## Participants

A total of 54 participants were recruited at the Chinese Academy of Sciences, Shanghai Branch. Twenty-six healthy participants (21 female, all right-handed, mean age = 24.0 ± 1.4 y) were recruited for Experiment 1. Two were excluded due to failure in completing the experiment or low conformity to task instructions, remaining 24 participants who completed the main experiment (19 female, mean age = 23.92 ± 1.41 y). Twenty-eight (22 female, mean age = 24.14 ± 1.51 y) participants were recruited for Experiment 2. Two quitted after behavioral training and two did not finish scanning due to technical problems with the scanner, resulting in 24 participants (20 female, mean age = 24.13 ± 1.60 y) in the final analyses. All participants were neurologically healthy and eligible for MRI, had normal or corrected-to-normal vision, provided written informed consent approved by the Ethics Committee of the Institute of Neuroscience, Center for Excellence in Brain Science and Intelligence Technology, Chinese Academy of Sciences (CEBSIT-2020028), and were monetarily compensated for their participation. Sample sizes were not estimated a priori but were comparable and even superior to those in previous studies.

## Stimuli and procedure

All stimuli were generated and presented using MATLAB (The MathWorks) and Psychtoolbox 3 extensions (*Brainard, 1997*; *Pelli, 1997*). During behavioral training, stimuli were presented on a ThinkVision monitor at a viewing distance of 45 cm. Behavioral responses were acquired with a keyboard. During scanning, stimuli were projected onto a SinoRad monitor (1280×1024 pixels, refreshing at 60 Hz) viewed through a coil-mounted mirror in the scanner at a viewing distance of 90.5 cm. Participants' behavioral responses were acquired with a Sinorad MRI-compatible button box.

### Behavioral training

In Experiment 1, prior to scanning, participants were trained to learn two novel rules, Rule A and Rule B, for categorizing orientations. Thirty oriented bars were used as sample stimuli, ranging from 5° to 179° (in increments of 6°; two participants used another set of thirty orientations ranging from 4° to 178°). Each abstract rule was constructed by two orthogonal boundaries that divided the thirty orientations into two categories with fifteen orientations each. Rule A and Rule B were orthogonal to each other. Corresponding boundaries were 20°/110° and 65°/155° (15°/105° and 60°/150° for the two participants using different stimuli sets).

Participants learned new rules through a rule-learning task. Each run of learning started with a rule disk informing the target rule. To avoid any potential verbal coding, rule specifics were visually illustrated as rule disks containing two distinct colors (colors randomly assigned to categories every time; see *Figure 1A* for an example). Rule disk was presented on the screen for 2 s followed by 1 s

of fixation. On each trial, an oriented bar (radius = 7°) was presented for 1 s followed by a delay of 1 s. Participants were instructed to report the category of the orientation by pressing a response key ('F' or 'J'). To avoid category-response mapping, we randomized the relationship between categories and key buttons across trials. Moreover, to avoid presenting rule boundaries explicitly, we presented key names at random positions within the range defining each category. In other words, participants had to memorize the exact rule boundaries as accurately as possible in order to find the correct key buttons for each trial. Feedback was given at the end of each trial to assist learning.

Participants completed 30 learning trials in each run. They reviewed rule disks after every 10 trials for memory reconsolidation. Each participant completed at least two runs for each rule. After achieving an average accuracy above 86% (26 out of 30 trials) for the first rule, they proceeded to learn the other rule and then to practice the main task for scanning (see next section). Learning order of rules was counterbalanced across subjects. Upon completion of practicing, participants needed to report the boundaries of learned rules as a qualification of behavioral training. If the total error in reported boundaries had exceeded 20°, participants had one more chance to learn by repeating the learning task. Two participants completed one additional behavioral training to recap rule knowledge prior to scanning.

Behavioral training for Experiment 2 followed the same procedure and used the identical sample set (30 orientations ranging from 4° to 178°, in increments of 6°) as Experiment 1, except that only one rule was trained. Half of the participants in Experiment 2 learned rule boundaries of 19°/109°; the other half learned rule boundaries of 67°/157°.

## Flexible WM task (fMRI task)

In Experiment 1, during scanning, participants completed a flexible WM task which implemented levels of control demand with different rules. To be specific, participants randomly switched between a maintenance task and a categorization task. In the maintenance task, participants needed to memorize stimulus information (i.e. orientations). In the categorization task, participants needed to categorize orientations following the rule that was randomly assigned and cued on a block basis. Procedure of the main task was visualized in *Figure 1A*. At the beginning of each block, participants were presented with a rule disk for 3 s, followed by a 2 s interval, instructing the categorization rule of the current block. For each trial, participants saw two oriented bars presented successively. Each bar was presented for 0.75 s, with an inter-stimulus-interval of 0.5 s. Sample sets were the same as those used in behavioral training. After a 0.5 s interval, a retro-cue occurred for 0.5 s, indicating the orientation of which participants should remember. After a 1.5 s delay, a task cue was displayed at fixation for 0.5 s, followed by an 8 s memory delay. The task cue was either a letter 'P' on maintenance trials, instructing participants to maintain the cued orientation during memory delay as precisely as possible; or the task cue was a letter 'C' on categorization trials, asking participants to categorize the cued orientation using the block rule during the delay. Then, participants were probed to respond within 2 s. On maintenance trials, participants needed to select the memorized orientation from two probe orientations; while on categorization trials, participants needed to report the category of the cued orientation. Response mapping followed the same operation as in the learning tasks. Inter-trial-intervals were randomly selected from 3, 5, and 7 s with an equal trial number, resulting in an average trial length of 20 s. Participants switched to the next block after every six categorization trials and three maintenance trials. Each run contained two blocks (i.e. 18 trials).

In Experiment 2, to isolate potential effect of rule switching, the categorization rule stayed the same throughout the experiment. In Experiment 2, participants randomly switched between the maintenance task and the categorization task. Each trial followed the same procedure as Experiment 1.

In Experiment 1, orientations, tasks, and cued target order (first or second) were counterbalanced across trials, resulting in an equal trial number of 90 across all three conditions (categorization-Rule A, categorization-Rule B, and maintenance). Nineteen out of the twenty-four participants completed 15 runs of the main task. One participant completed 13 runs due to technical difficulties with the scanner. Another four participants completed 30 runs across two scan sessions. The same counterbalancing procedure was conducted for Experiment 2 (90 trials for maintenance task and 180 trials for categorization task). In Experiment 2, six participants completed 15 runs of 18 trials; the other seven completed 18 runs of 15 trials each due to scanner limitations. At the end of scanning, participants reported the rule boundaries three times as a final check of their rule memory.

## Data acquisition

MRI data of Experiment 1 were collected using a 3 Tesla Siemens MRI scanner (Tim Trio; Siemens Healthineers) with a 32-channel head coil at the Functional Brain Imaging Platform at the Institute of Neuroscience, Center for Excellence in Brain Science and Intelligence Technology, Chinese Academy of Sciences. Functional scanning was performed using a gradient-echo echo-planar sequence with the following parameters: repetition time (TR)=1000 ms; echo time (TE)=30 ms; flip angle (FA)=40°; voxel size = 3 × 3 × 3 mm; multi-band accelerate factor = 4; matrix size = 74 × 74; slice number = 60. A high-resolution anatomical T1 image was collected before functional scanning (TR = 2300 ms; TE = 2.98 ms; FOV = 256 × 240 × 192 mm; voxel size = 1 × 1 × 1 mm). During scanning, participants' head positions were restricted with surrounding paddings to prevent head movements. MRI data of Experiment 2 were collected using identical procedures and settings except that the last eleven participants were scanned using a newly installed 3 Tesla Siemens MRI scanner (Prisma; Siemens Healthineers) at the Functional Brain Imaging Platform.

## Preprocessing

All preprocessing of individual MRI data was performed using AFNI (https://afni.nimh.nih.gov/) (*Cox, 1996*; *Cox and Hyde, 1997*). Functional data of all runs were registered to the last volume of the final run with the first eight volumes of each run removed. Then, individuals' aligned functional data were registered to their corresponding T1 volume. Alignment of registration was manually checked for each subject to ensure quality. The registered data were further motion corrected and detrended.

## ROI definition

Our primary ROI-based analyses focused on three most commonly-studied, WM-related brain areas: early visual cortex (EVC), intraparietal sulcus (IPS) in parietal cortex, and superior precentral sulcus (sPCS) in frontal cortex (*Ester et al., 2015*; *Hallenbeck et al., 2021*; *Yu and Shim, 2017*). We created anatomical ROI masks based on the probabilistic atlas by Wang and colleagues (*Wang et al., 2015*). EVC (merging bilateral V1, V2, and V3), IPS (merging bilateral IPS0-5), and sPCS (merging bilateral FEF) masks were generated by warping masks from the probabilistic atlas to individuals' anatomical image in their native space. In order to generate functional ROI masks, we then performed general linear models (GLMs) to quantify task-related univariate activity changes in each voxel. Task events were modeled using boxcar functions convolved with a canonical hemodynamic response function (durations of event epochs for sample, post retro-cue delay, memory delay, and response were 2.5 s, 2 s, 8.5 s, and 2 s, respectively). Six nuisance regressors were also included to account for head motion artifacts in the six dimensions of rigid body motion. Functional EVC mask was defined by the 500 most active voxels during sample display. Functional IPS and sPCS masks were defined by the 500 most active voxels during memory delay. Additional ROIs in the frontal cortex were defined using the HCP atlas (*Glasser et al., 2016*), including the inferior precentral sulcus (iPCS, generated by merging 6v, 6r, and PEF), inferior frontal sulcus (IFS, generated by merging IFJp, IFJa, IFSp, IFSa, and p47r), middle frontal gyrus (MFG, generated by merging 9-46d, 46, a9-46v, and p9-46v), and primary motor cortex (M1).

## MRI data analyses
### Multivariate inverted encoding modeling (IEM)

Neural representations of orientations were reconstructed using IEM (*Brouwer and Heeger, 2009*; *Brouwer and Heeger, 2011*; *Ester et al., 2015*; *Rademaker et al., 2019*; *Yu and Shim, 2017*) with custom MATLAB scripts on individuals' BOLD activation patterns in the ROIs. IEM provides an estimate of population-level reconstructions of stimulus-specific information. The general procedure for IEM includes using training data to train model weights and then applying weights to testing data to obtain reconstructed channel responses. For the main analyses, we used trials from all conditions to train and to test IEM in order to avoid potential biases from a specific task condition. Results for categorization task were averaged across rules for Experiment 1. Training and testing were performed for each TR separately. As a control, IEMs were also estimated for each condition separately (within-condition IEM). Training and testing underwent a leave-one-run-out cross-validation procedure, in which each run was taken out as the testing run, and the rest of the data served as the training run.

This procedure was iterated until all runs had served as training and testing runs. Results from all cross-validated folds were averaged. Detailed computations for each fold were elucidated below:

We first modeled responses of voxels into nine equidistant orientation channels (initial channels were 1°, 21°, 41°, 61°, 81°, 101°, 121°, 141°, 161°), characterizing voxel selectivity for orientations. At each channel, the modeled orientation tuning curve was a half-wave-rectified sinusoid raised to eighth power, defined as the function below ($c$ was the center of the channel):

$$\int (\theta) = \cos (\theta - c)^8$$

Population-level tuning responses of voxels were described using the function:

$$B_1 = WC_1$$

$B_1$ was the training dataset from our fMRI data ($v$ voxels $\times$ $n$ trials). $C_1$ represented the hypothesized channel responses ($k$ channels $\times$ $n$ trials) which were modulated by $W$, a weight matrix ($v$ voxels $\times$ $k$ channels).

The least-squared estimates of the weight matrix ($\hat{W}$) was computed using linear regression:

$$\hat{W} = B_1 C_1^T \left( C_1 C_1^T \right)^{-1}$$

The weight matrix was then applied to the test dataset to reconstruct estimated channel responses ($\hat{C}_2$):

$$\hat{C}_2 = \left( \hat{W}^T \hat{W} \right)^{-1} \hat{W}^T B_2$$

The analysis above was repeated for 20 times in step of 1° using leave-one-run-out cross-validation so that the nine channel centers covered all 180 orientations (*Rademaker et al., 2019*; *Yu and Postle, 2021*). All channel responses were combined to create responses for all 180 orientation channels. For statistical comparisons and for visualization, all channel responses were shifted to a common center of 0° (true orientation of trials). The responses from all trials were averaged to obtain reconstructed orientation representations for the test datasets.

To quantify the strength of each IEM reconstruction, we calculated reconstruction fidelity of channel responses (*Rademaker et al., 2019*). First, channel responses of each trial were shifted so that the orientation of each trial was centered in stimulus space. Thus, for each shifted channel response with a vector length of $r$ at orientation $\theta$, the orientation was wrapped onto a 2π circular space, and $r$ was projected to the vector at the true orientation (center of stimulus space, 0°) using the absolute angle between the channel and the center following the equation:

$$d = r\cos(abs(0^\circ - 2\theta))$$

The reconstruction fidelity was then calculated as the mean of all projected vectors $d$. A larger fidelity value indicates a stronger positive representation of orientation.

### Multivariate pattern analysis (MVPA)

Besides IEM, we tracked neural representations of stimulus and of category using linear Support Vector Machine (SVM) decoders. All decoding analyses were performed using the templateSVM and fitcecoc functions in MATLAB.

Decoding of stimulus was performed for every time point. We divided the thirty orientations into four bins of 45° each, two cardinal bins centered at 90° or 180° and two oblique bins centered at 45° or 135°. We then performed two two-way classifications, one trained and tested on cardinal bins, and the other trained and tested on oblique bins. We trained and tested decoders separately for each condition using the same leave-one-run-out cross-validation procedure as in IEM analyses. To avoid biases in model training, we randomly balanced the trial numbers for each bin in the training set. Decoding accuracies were then computed by averaging performance of cardinal and oblique classifiers. For the categorization task, we averaged accuracies across rules.

Decoding of category information for Experiment 1 was performed under each rule (90 trials for each rule) using a leave-one-trial-out cross-validation procedure (see next paragraph for details), and the decoding accuracies were then averaged across rules. Since Experiment 2 adopted a fixed rule with 180 trials in the categorization task, we randomly divided categorization trials into two halves with 90 trials each, and decoded category information for Experiment 2 using identical procedures as for Experiment 1.

Because closer orientations are more similar to each other inherently, orientations per se could contain categorical information by visual similarity. Thus, to isolate the influence of stimulus on category, in addition to the decoder using true category labels, we trained an opposite category decoder using category labels based on the opposite rule. If the two-way classification on categories only captured stimulus information, then true category and opposite category decoding should have had comparable performance. If abstract category information existed beyond stimulus information, then true category decoder should have outperformed the opposite category decoder. Thus, an abstract category index was calculated by subtracting opposite category decoding accuracy from true category decoding accuracy (i.e. chance level = 0). Since the opposite category decoding used re-assigned labels, to eliminate imbalance in trial number between true and opposite categories, we used a leave-one-trial-out cross-validation procedure for true category and opposite category decoders. Decoding for Experiment 1 was performed separately for each rule and were then averaged. Decoding for Experiment 2 was performed separately for randomly divided halves and averaged.

## Representational similarity analysis and linear mixed-effects modeling

We combined representational similarity analysis (RSA) with linear mixed-effects modeling (LMEM) to resolve contributions of stimulus and of category to the mixed neural representation. First, we constructed three hypothesized representational dissimilarity matrix (RDM) models (*Figure 5—figure supplement 2A*): a graded stimulus model (increasing distance between orientations as they move farther apart, corresponding to graded feature tuning responses), a discrete stimulus model (indicating equidistant orientation representations), and an abstract category model (0 for all orientations within the same category and 1 for different categories). Second, population-level neural RDMs were computed by calculating Pearson correlation distance (1 - correlation) on BOLD data between trials; distances were then averaged across time points within each of the 30 orientations and task, resulting in a 30 × 30 matrix for every task. Since there were two rules within individuals in Experiment 1 and across individuals in Experiment 2, to use orientation RDMs with the category RDMs simultaneously, we aligned the category RDMs between different categorization rules. In other words, we shifted the RDMs so that the first 15 rows and columns belonged to one category and the last 15 rows columns belonged to the other category. For instance, in Experiment 1, boundaries of Rule A were 20°/110°. The first 15 orientations in Rule A RDMs would be 22° - 106° and the other 15 orientations being 112° - 16°. With RDMs aligned, we then fitted neural RDMs in each ROI with the three model RDMs using LMEM to test the contribution of each model (graded stimulus, discrete stimulus or category). We used subject as the random effect. Significance of fixed effects was evaluated using p-values of the fixed effect coefficients.

## Recurrent neural network simulations

### RNN architecture

The network model was built following the details in previous work (*Masse et al., 2019*), and implemented in TensorFlow (version: Nvidia-tensorflow 1.15.0) (*Abadi et al., 2016*). The general network architecture consists of three layers of artificial units: the input, hidden, and output layers. The input layer contains units served to present various task-related signals corresponding to those in the fMRI paradigm, including orientations, retro-cues, task cues. In order to simulate neural activity patterns in hierarchically connected brain regions (EVC, IPS, and sPCS), we separated the hidden layer into three modules, each containing 200 recurrent units with short-term synaptic plasticity (STSP). Within each module, units were further divided into 80% excitatory and 20% inhibitory following Dale's principle. Similar to previous work (*Zhou et al., 2021*), modularity was achieved by constraining the recurrent connectivity in the hidden layer. Specifically, only posterior module's (module 1) excitatory units received inputs from the input layer and only anterior module's (module 3) excitatory units projected to the output units. Between-module connections were culled so that only 50% of a module's excitatory

units were randomly connected to their counterparts in the neighboring module(s), and vice versa (feedforward and feedback connections). Connections among inhibitory units remained strictly within-module in accordance with the observation that inhibitory connections in cortex are largely local. Thus, posterior, middle, and anterior modules were intended to simulate the three interconnected ROIs we used in the fMRI analyses that posited differently at the processing hierarchy. We specifically manipulated the output demand to investigate whether it would alter similarity of the results to the fMRI observations. To this end, one type of network architecture (RNN1) implemented a two-unit output layer with each unit corresponding to one of two possible response options, presented through the input units before the test period; In contrast, the other type of RNN architecture (RNN2) had additional units in the output layer, creating a demand for preserving the original stimulus information alongside categorical representations.

## Task simulation

Orientations were simulated as Gaussian signals from 15 orientation-tuned units in the input layer distributed equally across 0–180 degrees, forming a ring of receptive field. The magnitude of an orientation-tuned input unit represented the closeness of its preferred orientation to the input angle. Stimulus values were selected from an array of 20 orientations evenly spanned from (0 to 180) degrees. The sequentially-presented stimuli were presented through the same receptive field, followed by retro-cue and task cue indicated through the separate input units. For RNN1, before the test period when the network was required to make a choice, two response options were presented sequentially through the same input receptive field. The selection of the options varied slightly between the maintenance and categorization tasks: in maintenance, one orientation was always the cued sample while the other was randomly chosen from all other possible angles. In categorization, one option was taken from the same category as the cued sample but not necessarily the exact angle, while the other option was randomly chosen from orientations belonging to the other category. The network output (0,1) or (1,0) in the output units to report its choice. In comparison, RNN2 output (0,1) or (1,0) to report the category to which the cued orientation belonged in the categorization task, or (0,0) in the maintenance task. Importantly, the model also needed to report the cued orientation itself through a receptive field consisting of 15 orientation-tuned units in the output layer.

## Training parameters and procedure

The hyperparameters and procedure for training the models were consistent with those detailed in previous work (*Masse et al., 2019*), with the following exceptions: standard deviations of input and recurrent noise were set to 0.01 as our tasks were much harder to train compared to those used in the reference study (especially networks were trained on both tasks simultaneously). Lowering the noise level may provide an edge for the models to successfully learn to perform the tasks. In a similar vein, we also expanded the number of hidden units to 600 and number of training iteration to a maximum of 10000. Additionally, spike penalty was set to 0 for both RNN models to remove constraints on neuronal activity.

We trained 20 models for each type of RNN and results were obtained by averaging over all of them (for single-rule RNN, 10 models for each categorization rule). The goal of the training process was to minimize the mean square error between the model outputs and correct outputs during the test period via back propagation (with a 50 ms grace period at onset when model output was not taken into account in calculating error). Training was conducted in a block-interleaved fashion in which each gradient batch consisted of 300 maintenance, 300 categorization Rule A, and 300 categorization Rule B trials, with the task block order randomized (for single rule RNN, each batch consisted of 300 maintenance and 300 categorization trials). Training would automatically stop if the model achieved 90% accuracy in the last training batch on all tasks. For RNN2, the accuracies for category and stimulus outputs were calculated separately to ensure precision of the stimulus outputs.

## Population decoding

We measured the strength of stimulus and category representations through training SVMs on time-resolved neuronal activity. Activities were obtained by feeding new batches of tasks into the already successfully trained networks after freezing all connection weights to prevent further changes to the models' behaviors. The intrinsic noise for the recurrent layer was also set to 0 for decoding analyses.

To ensure accurate decoding results, we sampled large number of trials (900 trials for each condition) and implemented a five-fold cross-validation procedure in which 80% of trials were used as training set and the remaining 20% as testing set in each fold. Decoders were trained separately for each module and time point.

### Representational similarity analysis between RNN and human data

To quantify the similarity between RNN and human data, we conducted RSA between human and RNNs. First, for each RNN, we constructed RDMs for each module by calculating the Pearson correlation distance between the activation of each time point across trials and then averaging the distance across the delay period within each orientation and task. Human RDMs for early and late epochs were constructed using methods described in the previous section. Only orientations used in both human tasks and simulations were included for RSA. To prioritize the comparison of stimulus or category representation, we aligned the RDMs either by stimulus or category. For align-by-stimulus RDMs, RDMs were aligned to orientations, sorted from 1° to 180°, and in the order of maintenance and categorization tasks. For align-by-category RDMs, RDMs were aligned to categories, with the first half rows and columns under each task corresponding to one category and the other half corresponding to the other category. RDMs for different categorization rules were further averaged.

Next, for each module in each type of RNN (RNN1 vs. RNN2, flexible vs. fixed rule), we obtained a single model RDM by averaging the RDMs of all the 20 networks. Likewise, we obtained a single RDM for each ROI in each experiment for human data of individual participant. We then calculated Kendall's Tau correlation coefficients between human and model RDMs for each ROI and the corresponding module, and averaged the coefficient across participants. Significance of the correlation was determined using one-sided signed-rank test. Finally, to compared whether RNN1 or RNN2 was more similar to human patterns, we calculated the difference in Z-transformed coefficients between RNNs. Significance of difference was determined using sign-flip permutation with 1000 iterations (described below).

## Statistical testing

Participants' behavioral performance for the main task was assessed using accuracy and reaction time. Paired t-test was conducted for the two task types (maintenance & categorization) to evaluate differences between conditions.

Statistical significance was evaluated via a sign-flip permutation procedure for all other fMRI analyses. For example, to characterize the significance of IEM fidelity, we computed the p-value by comparing the true mean fidelity of our sample with a null distribution reflecting no IEM fidelity. The null distribution was created by randomly assigning 1 or –1 to fidelity scores of our sample and then averaging the sign-flipped samples for 10,000 times, resulting in a null distribution of 10,000 fidelity scores. To characterize the difference of IEM fidelity between tasks, we sign-flipped the fidelity sample for each condition and then averaged the difference for 10,000 times. The p-value was calculated by comparing the true mean difference with the generated null distribution of difference. p-values were corrected using FDR across ROIs, time , and tasks for all analyses unless specified. An early (5–10 s; 6th-11th TR) and late (11–16 s; 12th-17th TR) task epoch was also defined to facilitate comparisons between ROIs and experiments when needed.

For RNN decoding results, we pooled decoding accuracies across a critical task period (50–75 time points during delay) to produce summary statistics aligning with what was reported in the fMRI results. Average decoding results were corrected using the FDR method.

## Acknowledgements

We would like to thank Dr. Mu-ming Poo for valuable comments on an earlier version of the manuscript, and Dr. Tianming Yang for helpful discussions. This work was supported by the Strategic Priority Research Program of the Chinese Academy of Sciences (Grant No. XDB1010202), the Ministry of Science and Technology of China (STI2030-Major Projects 2021ZD0203701, 2021ZD0204202), the National Natural Science Foundation of China (32271089), CAS Project for Young Scientists in Basic Research (YSBR-071), and Shanghai Pujiang Program (22PJ1414400) to QY.

# Additional information

## Competing interests

Qing Yu: Reviewing editor, eLife. The other authors declare that no competing interests exist.

## Funding

| Funder | Grant reference number | Author |
| --- | --- | --- |
| Chinese Academy of Sciences | Strategic Priority Research Program of the Chinese Academy of Sciences, XDB1010202 | Qing Yu |
| Ministry of Science and Technology of the People's Republic of China | STI2030-Major Projects 2021ZD0203701 | Qing Yu |
| Ministry of Science and Technology of the People's Republic of China | STI2030-Major Projects 2021ZD0204202 | Qing Yu |
| National Natural Science Foundation of China | 32271089 | Qing Yu |
| Chinese Academy of Sciences | CAS Project for Young Scientists in Basic Research, YSBR-071 | Qing Yu |
| Science and Technology Commission of Shanghai Municipality | Shanghai Pujiang Program, 22PJ1414400 | Qing Yu |

The funders had no role in study design, data collection and interpretation, or the decision to submit the work for publication.

## Author contributions

Zhujun Shao, Data curation, Formal analysis, Investigation, Visualization, Methodology, Writing – original draft, Writing – review and editing; Mengya Zhang, Formal analysis, Methodology, Writing – original draft, Writing – review and editing; Qing Yu, Conceptualization, Supervision, Funding acquisition, Methodology, Writing – original draft, Writing – review and editing

## Author ORCIDs

Zhujun Shao ⓘ https://orcid.org/0009-0006-0886-4606
Mengya Zhang ⓘ https://orcid.org/0009-0006-9197-1698
Qing Yu ⓘ https://orcid.org/0000-0001-8480-7634

## Ethics

All participants provided written informed consent approved by the Ethics Committee of Institute of Neuroscience, Center for Excellence in Brain Science and Intelligence Technology, Chinese Academy of Sciences (CEBSIT-2020028).

Reviewer #1 (Public review): https://doi.org/10.7554/eLife.100287.4.sa1
Reviewer #2 (Public review): https://doi.org/10.7554/eLife.100287.4.sa2
Author response https://doi.org/10.7554/eLife.100287.4.sa3

# Additional files

## Supplementary files

MDAR checklist

## Data availability

The codes and data used to reproduce all figures can be found at Science Data Bank: https://doi.org/10.57760/sciencedb.21184.

The following dataset was generated:

| Author(s) | Year | Dataset title | Dataset URL | Database and Identifier |
|---|---|---|---|---|
| Shao Z, Zhang M, Yu Q | 2025 | Dataset and scripts for Stimulus representation in human frontal cortex supports flexible control in working memory | https://doi.org/10.57760/sciencedb.21184 | Science Data Bank, 10.57760/sciencedb.21184 |

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
