## [Editor Report · eLife Assessment]

This work presents **important** findings that the human frontal cortex is involved in a flexible, dual role in both maintaining information in short-term memory, and controlling this memory content to guide adaptive behavior and decisions. The evidence supporting the conclusions is **compelling**, with a well-designed task, best-practice decoding methods, and careful control analyses. The work will be of broad interest to cognitive neuroscience researchers working on working memory and cognitive control.

---

## [Referee Report · Reviewer #1 (Public review)]

Summary:

In this manuscript, Shao et al. investigate the contribution of different cortical areas to working memory maintenance and control processes, an important topic involving different ideas about how the human brain represents and uses information when no longer available to sensory systems. In two fMRI experiments, they demonstrate that human frontal cortex (area sPCS) represents stimulus (orientation) information both during typical maintenance, but even more so when a categorical response demand is present. That is, when participants have to apply an added level of decision control to the WM stimulus, sPCS areas encode stimulus information more than conditions without this added demand. These effects are then expanded upon using multi-area neural network models, recapitulating the empirical gradient of memory vs control effects from visual to parietal and frontal cortices. Multiple experiments and analysis frameworks provide support for the authors' conclusions, and control experiments and analysis are provided to help interpret and isolate the frontal cortex effect of interest. While some alternative explanations/theories may explain the roles of frontal cortex in this study and experiments, important additional analyses have been added that help ensure a strong level of support for these results and interpretations.

Strengths:

- The authors use an interesting and clever task design across two fMRI experiments that is able to parse out contributions of WM maintenance alone along with categorical, rule-based decisions. Importantly, the second experiments only uses one fixed rule, providing both an internal replication of Experiment 1's effects and extending them to a different situation when rule switching effects are not involved across mini-blocks.

- The reported analyses using both inverted encoding models (IEM) and decoders (SVM) demonstrate the stimulus reconstruction effects across different methods, which may be sensitive to different aspects of the relationship between patterns of brain activity and the experimental stimuli.

- Linking the multivariate activity patterns to memory behavior is critical in thinking about the potential differential roles of cortical areas in sub-serving successful working memory. Figure 3's nicely shows a similar interaction to that of Figure 2 in the role of sPCS in the categorization vs. maintenance tasks. This is an important contribution to the field when we consider how a distributed set of interacting cortical areas support successful working memory behavior.

- The cross-decoding analysis in Figure 4 is a clever and interesting way to parse out how stimulus and rule/category information may be intertwined, which would have been one of the foremost potential questions or analyses requested by careful readers.

- Additional ROI analyses in more anterior regions of the PFC help to contextualize the main effects of interest in the sPCS (and no effect in the inferior frontal areas, which are also retinotopic, adds specificity). And, more explanation for how motor areas or preparation are likely not involved strengthens the takeaways of the study (M1 control analysis).

- Quantitative link via RDM-style analyses between the RNNs constructed and fMRI data.

Weaknesses:

- In the given tasks, multiple types of information codes may be present, and more detail on this possibility could always be added analytically or in discussion. However, the authors have added beneficial support to this comparison in this version of the manuscript.

- The space of possible RNN architectures and their biological feasibility could always be explored more, but links between the fMRI and RNN data provide a good foundation for this work moving forward.

---

## [Referee Report · Reviewer #2 (Public review)]

Summary:

The author provide evidence that helps resolve long-standing questions about the differential involvement of frontal and posterior cortex in working memory. They show that whereas early visual cortex shows stronger decoding of memory content in a memorization task vs a more complex categorization task, frontal cortex shows stronger decoding during categorization tasks than memorization tasks. They find that task-optimized RNNs trained to reproduce the memorized orientations show some similarities in neural decoding to people. Together, this paper presents interesting evidence for differential responsibilities of brain areas in working memory.

Strengths:

This paper was overall strong. It had a well-designed task, best-practice decoding methods, and careful control analyses. The neural network modeling adds additional insight into the potential computational roles of different regions.

Weaknesses:

Few. The RNN-fMRI correspondence could be a little more comprehensive, but the paper contributes a compelling set of empirical findings and interpretations that can inform future research.

---

## [Author Response]

The following is the authors’ response to the previous reviews.

We would like to sincerely thank the reviewers again for their insightful comments on the previous version of our manuscript. In the last round of review, the reviewers were mostly satisfied with our revision but raised a few suggestions and/or remaining concerns. We have further edited the manuscript to address these concerns.

**Reviewer #1:**
- An explicit, quantitative link between the RNN and fMRI data is perhaps a last point that would integrate the RNN conclusion and analyses in line with the human imaging data.
**Reviewer #2:**
- Few. While more could be perhaps done to understand the RNN-fMRI correspondence, the paper contributes a compelling set of empirical findings and interpretations that can inform future research.

To better align the RNN and fMRI results qualitatively, we performed an additional representational similarity analysis (RSA) on the data. Specifically, we computed the representational dissimilarity matrices (RDMs) for fMRI and RNN data separately, and calculated the correlation between the RDMs to quantify the similarity between fMRI data and different RNN models. We found that, consistent with our main claims, RNN2 generally demonstrated higher similarity with the fMRI data compared to RNN1. These results provide further support that RNN2 aligns better with human neuroimaging data. We have included this result (lines 496-505) and the corresponding figure (Figure 7) in the manuscript.

**Reviewer #1:**
- As Rev 2 mentions, multiple types of information codes may be present, and the response letter Figure 5 using representational similarity (RSA) gets at this question. It would strengthen the work to, at minimum, include this analysis as an extended or supplemental figure.

Following this suggestion, we have now included Response Letter Figure 5 from the previous round of review in the manuscript (lines 381-387 and Appendix 1 – figure 7).

**Reviewer #1:**
- To sum up the results, a possible, brief schematic of each cortical area analyzed and its contribution to information coding in WM and successful subsequent behavior may help readers take away important conclusions of the cortical circuitry involved.

Following this suggestion, we have added a schematic figure illustrating the contribution of each cortical region in our experiment to better summarize our findings (Figure 8).

We hope that these changes further clarify the issues and strengthen the key claims in our manuscript.